# Universal Graph Convolutional Networks

**Di Jin[1†], Zhizhi Yu[1†], Cuiying Huo[1], Rui Wang[1], Xiao Wang[2*], Dongxiao He[1], and Jiawei Han[3]**
[1]College of Intelligence and Computing, Tianjin University, Tianjin, China
[2]School of Computer Science (National Pilot Software Engineering School),
Beijing University of Posts and Telecommunications, Beijing, China
[3]Department of Computer Science, University of Illinois at Urbana-Champaign, Champaign, IL, USA
{jindi, yuzhizhi, huocuiying, wr1895, hedongxiao}@tju.edu.cn
xiaowang@bupt.edu.cn, hanj@illinois.edu

## Abstract

Graph Convolutional Networks (GCNs), aiming to obtain the representation of a node by aggregating its neighbors, have demonstrated great power in tackling various analytics tasks on graph (network) data. The remarkable performance of GCNs typically relies on the homophily assumption of networks, while such assumption cannot always be satisfied, since the heterophily or randomness are also widespread in real-world. This gives rise to one fundamental question: whether networks with different structural properties should adopt different propagation mechanisms? In this paper, we first conduct an experimental investigation. Surprisingly, we discover that there are actually segmentation rules for the propagation mechanism, i.e., 1-hop, 2-hop and $k$-nearest neighbor ($k$NN) neighbors are more suitable as neighborhoods of network with complete homophily, complete heterophily and randomness, respectively. However, the real-world networks are complex, and may present diverse structural properties, e.g., the network dominated by homophily may contain a small amount of randomness. So can we reasonably utilize these segmentation rules to design a universal propagation mechanism independent of the network structural assumption? To tackle this challenge, we develop a new universal GCN framework, namely U-GCN. It first introduces a multi-type convolution to extract information from 1-hop, 2-hop and $k$NN networks simultaneously, and then designs a discriminative aggregation to sufficiently fuse them aiming to given learning objectives. Extensive experiments demonstrate the superiority of U-GCN over state-of-the-arts. The code and data are available at https://github.com/jindi-tju.

## 1 Introduction

Real-world complex systems can often be viewed as networks, such as social networks, biological networks and citation networks. Recently, research of analyzing networks with deep learning has received widespread attention both in academia and industry. In particular, Graph Convolutional Networks (GCNs) [14], which obtain the meaningful representation of nodes in the network by integrating the neighborhood information, have achieved great success and been widely applied in tackling network analytics tasks, such as node classification [23, 28], link prediction [33] and recommendation [30, 17].

While the success of GCNs and their variants [1, 6], a key weakness is the homophily assumption of networks, which restricts their performance on general network data. To be specific, most GCNs

---

†Equal contributions.
*Corresponding author.

seem to be tailor-made to work on homophily networks [18], where nodes within the same class tend to connect with each other. In fact, heterophily [20] networks, where nodes of different classes tend to link together, are also widespread in real-world. For example, different types of amino acids are more likely to be connected in protein structure [35], and fraudsters tend to connect to accomplices than to other fraudsters in transaction networks [21]. Furthermore, the random networks also often exist in real-world, such as railway networks, where the edges between nodes are more likely to be randomly generated.

Most popular GCNs typically obtain node embeddings of these networks using 1-hop network neighbors as neighborhoods for information propagation [12, 29]. However, considering the different structural properties of networks (e.g., homophily or heterophily), whether different networks should adopt a unified or different propagation mechanisms? This is a very important question for GCNs since they mainly gain better performance through the propagation of information. A well informed answer can help us better understand the essence of GCNs, such as how different types of nodes affect the propagation, and what type of nodes are really required to achieve a certain level of predictive accuracy aiming to different networks.

Several recent works have studied the networks with different structural properties. For example, Pei *et al*. [22] consider the heterophlily property of networks, and propose a geometric aggregation scheme to overcome neighborhood structural information losing and long-range dependencies lacking. Zhu *et al*. [35] design an effective model which improves the representation power of GCNs under heterophily through theoretical and empirical analysis. Chien *et al*. [4] introduce a new generalized pageRank (GPR) architecture to jointly optimize node feature and topological information extraction. Bo *et al*. [2] assess the roles of low-frequency and high-frequency signals, and propose an efficient method that can adaptively integrate different signals in the process of message passing. However, there is still a lack of insightful understanding from the perspective of propagation mechanism.

As the first contribution of this study, we conduct experiments analysing the propagation mechanism of GCNs in networks with different structural properties. Surprisingly, our experiments clearly illustrate that for networks with complete homophily, complete heterophily and randomness, 1-hop, 2-hop and $k$-nearest neighbor ($k$NN) neighbors are more suitable as neighborhoods for information propagation, respectively. This means that the depicting ability of the current propagation mechanism of GCNs is limited, and networks with different structural properties may need to adopt different propagation mechanisms.

In fact, while these segmentation rules seem to be able to select appropriate nodes as neighborhoods in an ideal way, the real-world networks are complex, and may present diverse properties, e.g., the network dominated by homophily may contain a small amount of randomness or heterophily. A natural question is, "*Can we reasonably utilize these segmentation rules to design a universal propagation mechanism independent of the network structural assumption*?"

To tackle this challenge, we propose a novel and universal GCN model, i.e., U-GCN, for general network data. The central idea is that we learn node embeddings by making full use of the information from 1-hop, 2-hop and $k$NN neighbors, and fuse them adaptively to derive deeper correlation information for the given learning objectives. To be specific, we first introduce a multi-type convolution mechanism. It uses 1-hop network (i.e., original input network), 2-hop network and $k$NN network that constructed by 1-hop, 2-hop and $k$NN neighbors for direct information propagation separately, and utilizes a node-level attention mechanism for each network, to extract three specific embeddings. We then make a discriminative aggregation to learn out the importance of these three embeddings, thereby extracting the most correlated information aiming to the ground truth such as node classification. Extensive experiments on a series of benchmark datasets demonstrate the superiority of U-GCN over some state-of-the-arts.

## 2 Notations and Preliminaries

Let $G = (A, X)$ be an undirected attributed network, where $A \in \mathbb{R}^{n \times n}$ represents the symmetric adjacency matrix with $n$ nodes, and $X \in \mathbb{R}^{n \times p}$ is the attribute (content) matrix of $p$ attributes per node. Concretely, $a_{ij} = 1$ denotes there is an edge between nodes $v_i$ and $v_j$, or 0 otherwise; and $x_i$ represents the attribute vectors of node $v_i$.

Given an attribute network $G$, and a labeled node set $V_L$ containing $u \ll |V|$ nodes, where each node $v_i \in V_L$ contains a unique class label $y_i \in Y$. The goal of semi-supervised node classification is to infer the labels of nodes in $V \backslash V_L$ by learning a classification function $\mathcal{F}$.

**Homophily.** In this work, the level of homophily ratio of edges [35] is used to define networks with strong homophily/heterophily. Specifically, the level of homophily ratio of edges is the fraction of edges in a network which connect nodes that have the same class label (i.e., intra-class edges), described by:

$$\alpha = \frac{|(v_i, v_j) : a_{ij} = 1 \wedge y_i = y_j|}{|m|}, \tag{1}$$

where $m$ is the number of edges. Networks with $\alpha$ closer to 1 tend to have more edges connecting nodes within the same class, or stronger homophily; whereas networks with $\alpha$ closer to 0 have more edges connecting nodes in different classes, or stronger heterophily.

**Graph Convolutional Network.** Graph Convolutional Network (GCN) [14] is a variant of multi-layer convolutional neural networks that operates directly on networks. It learns embedding of each node by iteratively aggregating the information from its neighbors. Mathematically, let $H^{(l)}$ be the feature representation of the $l$-th layer, and $H^{(0)}$ be the node attribute matrix, the forward propagation can be defined as:

$$H^{(l)} = \sigma(\tilde{D}^{-\frac{1}{2}} \tilde{A} \tilde{D}^{-\frac{1}{2}} H^{(l-1)} W^{(l)}), \tag{2}$$

where $\tilde{A} = A + I$ stands for the adjacency matrix with self-loops, $\tilde{D}$ the node degree matrix of $\tilde{A}$, i.e., $\tilde{D}_{ii} = \sum_j \tilde{A}_{ij}$, $W^{(l)}$ a trainable weight matrix and $\sigma$ the non-linear activation function. While GCN works well on several network analysis tasks such as node classification [10, 15], it still has a fundamental problem, that is, homophily assumption of networks, which leads to the main contribution in this work, i.e., analyse what type of nodes are more suitable as neighborhoods for direct information propagation independent of the network structural assumption.

## 3 Motivating Observations

Here, we present a simple yet intuitive case study to illustrate and analyze the performance of GCN changes with different propagation mechanisms. The main idea is that we will apply GCN to networks with different structural properties utilizing three types of nodes: 1-hop, 2-hop and $k$-nearest neighbor ($k$NN) neighbors, which are often believed to be the effective neighborhoods for node classification in networks [28, 35], to realize the information propagation, respectively. Then, we will check the performance of GCN on these cases. A universal propagation mechanism should provide a good result in general network data. However, if the performance drops sharply in comparison with the other two situations, this will demonstrate that networks with different structural properties may need to use different propagation mechanisms.

**Setup.** We conduct experiments on the Newman artificial networks [7] with different properties. The network consists 128 nodes divided into 4 classes, where each node has on average $z_{in}$ edges (i.e., intra-class edges) connecting to nodes of the same class and $z_{out}$ edges (i.e., inter-class edges) to nodes of other classes, and $z_{in} + z_{out} = 16$. Note that here we utilize two indicators: $\rho_{in} = z_{in}/32$ and $\rho_{out} = z_{out}/96$, to determine the network property, i.e., $\rho_{in} > \rho_{out}$, $\rho_{in} = \rho_{out}$ and $\rho_{in} < \rho_{out}$ means the network with homophily, randomness and heterophily, respectively.

For node attributes, we generate a $4h$-dimensional binary attributes (i.e., $x_i$) for each node to form 4 attribute clusters, corresponding to the 4 classes [9]. To be specific, for every node in the $m$-th class, we use a binomial distribution with mean $p_{in} = h_{in}/h$

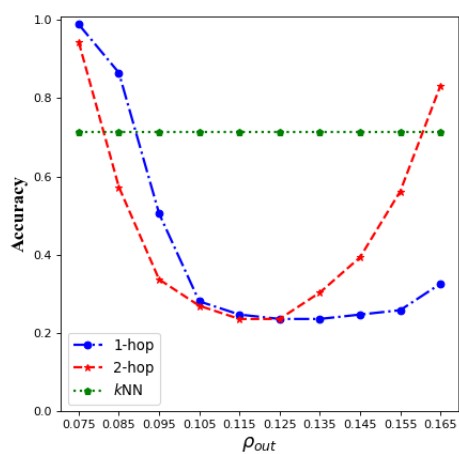

Figure 1: The performance of GCN of using different propagation mechanisms: 1-hop, 2-hop and $k$NN neighbors as neighborhoods respectively on Newman networks.

to generate a $h$-dimensional binary vector as its $((m-1) \times h + 1)$-th to $(m \times h)$-th attributes, and generated the rest attributes using a binomial distribution with mean $p_{out} = h_{out}/(3h)$. In our experiments, we set $4h = 200$ and $h_{out} = 4$ $(h_{in} + h_{out} = 16)$, so that $p_{in} > p_{out}$, the $h$-dimensional attributes are associated with the $m$-th class with a higher probability, whereas the rest $3h$ attributes are irrelevant.

As shown in Figure 1, for networks with strong homophily (e.g., $\rho_{out} = 0.075$), it is easy to obtain high accuracy using 1-hop network neighbors. However, as the inter-class edges increase, the accuracy is rapidly reduced. This mainly due to the homophily assumption, preventing GCN from effectively fusing information. On the other hand, for networks with strong heterophily (e.g., $\rho_{out} = 0.165$), it is surprising that, the accuracy of GCN of using 2-hop neighbors as neighborhoods (i.e., 83.15%) is much higher than that of using 1-hop network neighbors (i.e., 32.85%). Since the homophily ratio of 2-hop neighbors may rise with the increase of inter-class edges, GCN of using 2-hop neighbors is more effective to some extent. Interestingly, we can find that GCN of utilizing $k$NN is easy to get the staple accuracy, i.e., 71.46%. In particular, it is much higher than those of using 1-hop and 2-hop neighbors on complete random network (i.e., $\rho_{out} = 0.125$).

**Summary.** This case study shows that the current propagation mechanism of GCN is not universal for general network data, but we can find that there are rules in several special situations (i.e., complete random network). This motivates us that networks with different structural properties may need adopt different propagation mechanisms.

We conduct extra experiments on Newman networks [7] with complete homophily, randomness and complete heterophily utilizing GCN, so as to discover more appropriate propagation mechanism to select valuable nodes as neighborhoods, and thus improving the performance of GCNs for different networks. One straightforward strategy is to learn network embeddings with different GCNs using different types of nodes (i.e., 1-hop, 2-hop and $k$NN neighbors), and concatenate the embeddings into a single vector, so as to use the discriminative aggregation mechanism (which will be introduced in Section 4.2 below) to learn their importance for node classification. We show the attention values as a function of the number of training iterations in Figure 2.

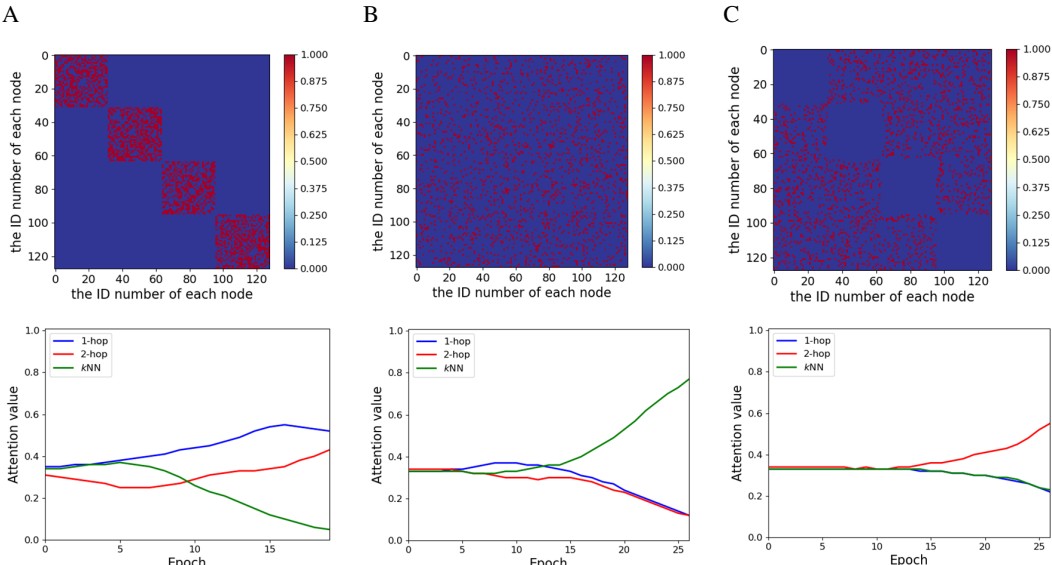

Figure 2: An example illustrating that the importance of three different types of neighbors (i.e., 1-hop, 2-hop and $k$NN neighbors) changes with network properties. The upper part of A-C represents networks with complete homophily, randomness and complete heterophily, respectively; while the lower part denotes the attention values as a function of the number of training iterations in corresponding networks.

***Observation 1:*** *Network with complete homophily tends to obtain better performance utilizing 1-hop network neighbors for direct information propagation.* For the network in Figure 2A, the edges exist only in nodes within the same class, or complete homophily. As shown, 1-hop network neighbors

show great importance as the increase of training iterations. This partly validates that under the setting of complete homophily, 1-hop neighbors are more effective for direct information propagation compared to 2-hop and $k$NN neighbors.

***Observation 2:*** *Network with randomness tends to get better performance utilizing kNN for direct information propagation.* The network in Figure 2B exhibits more randomness than Figure 2A, that is, complete random network. Obviously, with the increase of training iterations, the attention value of $k$NN is much higher than those of 1-hop and 2-hop neighbors. Since $k$NN typically constructed according to the similarity of node attributes, it can still realize the information fusion effectively, compared with the other two types of neighbors, in case that the network topology contains noise.

***Observation 3:*** *Network with complete heterophily tends to obtain better performance utilizing 2-hop neighbors for direct information propagation.* For the network in Figure 2C, the edges exist only in nodes within different classes, or complete heterophily. While the learned attention values of these three types of neighbors differ slightly, the importance of 2-hop neighbors is relatively higher. This is mainly due to the fact that the homophily ratio of 2-hop neighbors becomes higher with the increase of inter-class edges (which is often the real life in many network analysis tasks).

While networks with different structural properties provide better performance utilize different propagation mechanisms, the real-world networks are complex, and may show diverse properties, e.g., the network dominated by homophily may contain a small amount of heterophily. Therefore, it is imperative to explore a universal propagation mechanism for GCNs independent of the network structural assumption.

**Theorem 1.** The real-world networks can be approximately decomposed into a mixture of three kinds of simple networks, namely complete homophily, complete random and complete heterophily, in different proportions.

*Proof.* Given a network $G$, where $n$ and $m$ denote the number of nodes and edges, respectively. Assuming that $r$ edges ($0 \leq r \leq m$) are generated randomly, i.e., the probability of nodes connecting nodes within the same class or different classes is the same, which form a complete random network (with $n$ nodes and $r$ edges). The remaining edges connecting two nodes within the same class can then be regarded as composing a complete homophily network, or a complete heterophily network.

**Summary.** Now, we can conclude that a universal GCN model may not only consider the 1-hop (Observation 1), but also the 2-hop (Observation 2) and $k$NN neighbors (Observation 3) for direct information propagation. More importantly, considering different network properties can be more correlated with one of them or even their combinations, the model itself should adaptively learn their corresponding importance, so as to achieve feature fusion more effectively. This case study, although leveraging specific artificial networks, is representative because real-world networks can often be considered as the combination of these three simple network cases.

## 4  Our Proposed Approach

To address the homophily assumption of GCNs, our basic idea is to design a universal GCN framework which is suitable for general networks with any structural properties. It can not only make full use of the information from 1-hop, 2-hop and $k$NN neighbors, but also fuse them sufficiently aiming to given learning objectives. In this section, we start by proposing a new simple multi-type convolution mechanism over three kinds of neighbors, and then introduce a discriminative aggregation to learn the importance of each part: 1-hop, 2-hop and $k$NN neighbors, automatically.

### 4.1  Multi-type Convolution Mechanism

To capture the information from 2-hop and $k$NN neighbors, we construct a 2-hop network $G_R = (A_R, X)$ based on original input network $G_D = (A_D, X)$, and a $k$NN network $G_F = (A_F, X)$ based on node feature matrix $X$.

**2-hop Network.** For adjacency matrix $A_R$, considering that the number of neighbors at exactly 2 hops away may raise exponentially with the increase of network scale, we introduce a constraint, i.e., select node pairs connected by at least two different paths for each node to set edges. Simultaneously, we adopt the classic two-layers GCN to perform message passing on this 2-hop network $(A_R, X)$,

and the $l$-th layer embedding matrix $H_R^{(l)}$ can be denoted as:

$$H_R^{(l)} = \sigma(\tilde{D}_R^{-\frac{1}{2}} \tilde{A}_R \tilde{D}_R^{-\frac{1}{2}} H_R^{(l-1)} W_R^{(l)}), \tag{3}$$

where $\tilde{A}_R = A_R + I$, $I$ is the identity matrix, $\tilde{D}_R$ the diagonal degree matrix of $\tilde{A}_R$, $W_R^{(l)}$ the weight matrix and $\sigma$ the non-linear activation function such as ReLU or Sigmoid. In this way, we can learn the node embeddings that capture the specific information from 2-hop neighbors.

$k$**NN Network.** There are many ways to obtain $k$NN for each node, such as Jaccard similarity, Cosine similarity and Gauss kernel. In what follows, we calculate the similarity matrix $S \in \mathbb{R}^{n \times n}$ among $n$ nodes utilizing Cosine similarity, which adopts the cosine value of the angle between two vectors to measure the similarity. Mathematically, let $x_i$ be the feature vectors of node $v_i$, the similarity $s_{ij}$ between nodes $v_i$ and $v_j$ is defined as:

$$s_{ij} = \frac{x_i \cdot x_j}{|x_i||x_j|}. \tag{4}$$

Then, the adjacency matrix $A_F$ can be obtained by choosing top $k$ similar node pairs for each node to set edges. Accordingly, the $l$-th layer embedding matrix $H_F^{(l)}$ that gains the information from $k$NN can be calculated in the same way as in 2-hop network. Also of note, we use a linear algorithm Ball-tree [16] for the calculation of $k$NN network which will not increase the complexity of GCN.

As for 1-hop network neighbors, we acquire the $l$-th layer embedding matrix $H_D^{(l)}$ performing direct information propagation on original input network $G_D = (A_D, X)$. Therefore, the specific information encoded in 1-hop network neighbors can be extracted.

**Node-level Attention.** Before aggregating the information from original input network, 2-hop network and $k$NN network, we should note that the network-based neighbors of each node contribute to the embedding of the target node in different degrees. Here we adopt node-level attention [26] to learn the importance of network-based neighbors for each node. To be specific, given a node pair $(v_i, v_j)$ and a specified network type $t$ (where $t \in \{G_D, G_R, G_F\}$), the importance coefficient between nodes $v_i$ and $v_j$ can be formulated as:

$$e_{ij}^t = \text{LeakyReLU}(\mu_t^T [Wh_i || Wh_j]), \; \alpha_{ij}^t = \text{softmax}_j(e_{ij}^t) = \frac{\exp(e_{ij}^t)}{\sum_{r \in N_i^t} \exp(e_{ir}^t)}, \tag{5}$$

where $\mu_t$ is the parameterized attention vector for network type $t$, and $W$ the mapping matrix applied to each node. Then, the embedding of node $v_i$ for network type $t$ can be aggregated by the neighbor's embeddings with its corresponding weight coefficients as:

$$h_i^t = \sigma(\sum_{j \in N_i^t} \alpha_{ij}^t Wh_j). \tag{6}$$

### 4.2  Discriminative Aggregation

After the multi-type convolution above, we then perform a discriminative aggregation utilizing the attention mechanism, so as to learn the contributions of 1-hop, 2-hop and $k$NN neighbors automatically based on the given learning objectives. To be specific, for each node $v_i$, let $h_i^t$ denote its embedding in $H_t$, the attention value $\beta_i^t$ can then be represented as:

$$\beta_i^t = q^T \cdot \tanh(W_t \cdot (h_i^t)^T + b_t), \tag{7}$$

where $q$ denotes the parameterized attention vector, $W_t$ the weight matrix and $b_t$ the bias vector.

After obtaining the attention value of each network, i.e., $\beta_i^D$, $\beta_i^R$, $\beta_i^F$, we normalize them via softmax function to get the final weight:

$$\gamma_i^t = \text{softmax}(\beta_i^t) = \frac{\exp(\beta_i^t)}{\sum_t \exp(\beta_i^t)}. \tag{8}$$

Obviously, a larger $\gamma_i^t$ value means that the corresponding embedding is more important. The ouput embedding $H$ can then be aggregated by these network-specific embeddings with its corresponding weight coefficients as:

$$H = \sum_t \gamma_i^t \cdot H^t. \tag{9}$$

Following GCN, we define the loss function by using cross entropy as:

$$\mathcal{L} = -\sum_{i \in \mathcal{Y}_L} \sum_{f=1}^{F} Y_{if} \ln H_{if}, \qquad (10)$$

where $\mathcal{Y}_L$ is the set of node indices that have labels, $Y$ the label indicator matrix, and $F$ the dimension of the output embedding, which is equal to the number of classes.

## 5  Experiments

We first give the experimental setup, and then compare our U-GCN with some state-of-the-arts on node classification. We finally give an in-depth analysis of different components of our new approach.

### 5.1  Experimental Settings

**Datasets.** We adopt eight public network datasets with edge homophily ratio $\alpha$ ranging from strong homophily to strong heterophily, as shown in Table 1, to evaluate the performance of different methods. We use three citation networks Cora, CiteSeer and PubMed [19, 25], two Wikipedia networks Chameleon and Squirrel [24], and three webpage networks[1] Cornell, Wisconsin and Texas.

**Baselines.** We compare our U-GCN with eight baselines: (1) the methods utilizing both topological and attribute information: GCN [14], GAT [26], GraphSAGE [8], JK-Net [29], SSP [11], Geom-GCN [22] and GCN-LPA [27], and (2) the method using node attribute: MLP. Especially, GCN is the base of our U-GCN.

**Parameter Settings.** For all methods, we set the dropout rate to 0.6 and use the same splits for training, validation and testing sets. We run 5 times with the same partition and report the average results. We employ the Adam optimizer with the learning rate setting to 0.005 and apply early stopping with a patience of 20. In addition, we set the number of attention heads to 8, weight decay $\in \{5e-3, 5e-4\}$, and $k \in \{3...7\}$ for $k$-nearest neighbor network.

Table 1: Dataset Statistics.

| Datasets | Cora | Pubm. | Cite. | Corn. | Cham. | Squi. | Wisc. | Texa. |
|---|---|---|---|---|---|---|---|---|
| #Nodes | 2708 | 19717 | 3327 | 183 | 2277 | 5201 | 251 | 183 |
| #Edges | 5429 | 44338 | 4732 | 298 | 36101 | 217073 | 515 | 325 |
| #Features | 1433 | 500 | 3703 | 1703 | 2325 | 2089 | 1703 | 1703 |
| #Classes | 7 | 3 | 6 | 5 | 5 | 5 | 5 | 5 |
| $\alpha$ | 0.83 | 0.79 | 0.71 | 0.30 | 0.25 | 0.22 | 0.16 | 0.06 |

### 5.2  Node Classfication

On the node classification task, we use accuracy as the evaluation metric, and the relevant results are summarized in Table 2. As shown, we observe that the U-GCN has consistently strong performance across the full spectrum of high-middle-low homophily. To be specific, on the dataset with strong homophily, e.g., Cora and Citeseer, U-GCN is comparable with the best baseline GAT that based on homophily assumption. On the dataset with middle homophily, e.g., Cornell and Chameleon, U-GCN is 3.88% and 1.38% more accurate than the best baselines MLP and GCN-LPA, respectively. Above all, on the dataset with strong heterophily, e.g., Wisconsin and Texas, our model U-GCN outperforms the best baseline MLP by a very large margin, i.e., 5.69% and 5.83%, which has been proved to be superior to a number of existing GNNs at the low level of homophily [35]. These results not only demonstrate the superiority of the new multi-type convolution mechanism that makes full use of the information from 1-hop, 2-hop and $k$NN neighbors, but also validate the effectiveness for distinguishing importance of information from different propagation mechanisms. In addition, the performance of U-GCN is much better than that of GCN, which further demonstrates the effectiveness of designing a universal propagation mechanism independent of network structural assumption.

---

[1]http://www.cs.cmu.edu/afs/cs.cmu.edu/project/theo-11/www/wwkb

Table 2: Comparisons on node classification (Percent).

| Methods | Cora | Pubm. | Cite. | Corn. | Cham. | Squi. | Wisc. | Texa. |
|---|---|---|---|---|---|---|---|---|
| GCN | 82.93 | 83.29 | 73.12 | 46.51 | 52.32 | 33.10 | 47.73 | 52.71 |
| GAT | 83.13 | 84.42 | 72.04 | 48.06 | 51.38 | 32.27 | 46.59 | 49.61 |
| SSP | 81.08 | 79.50 | 71.13 | 55.04 | 21.87 | 19.72 | 49.37 | 55.04 |
| JK-Net | 81.27 | **86.15** | 71.74 | 52.71 | 53.95 | 33.51 | 48.30 | 51.94 |
| GraphSage | 82.20 | 83.03 | 71.41 | 53.49 | 42.29 | 26.89 | 56.82 | 53.49 |
| Geom-GCN | 74.27 | 83.49 | 73.79 | 54.26 | 38.66 | 32.22 | 53.41 | 64.34 |
| GCN-LPA | 82.33 | 85.83 | 72.29 | 49.61 | 52.69 | 33.48 | 50.57 | 48.84 |
| MLP | 63.33 | 83.08 | 67.74 | 65.89 | 41.35 | 29.44 | 64.20 | 65.89 |
| U-GCN | **84.00** | 85.22 | **74.08** | **69.77** | **54.07** | **34.39** | **69.89** | **71.72** |

## 5.3 Ablation Study

Similar to most deep learning models, U-GCN also contains some important components that may have significant impact on the performance. To gain deeper insight into the contributions of different components involved in our approach, we conduct experiments on comparing U-GCN with four variations. The variants are as follows: 1) GCN which serves as the base framework of U-GCN of using 1-hop network neighbors for propagation, 2) GCN of employing 2-hop neighbors for direct propagation, named as U-GCN-1, 3) GCN of utilizing $k$NN for direct propagation, named as U-GCN-2, and 4) U-GCN of removing 2-hop neighbors, named as U-GCN-3. We take their comparison on node classification as an example.

As shown in Table 3, compared to GCN, U-GCN-1 (and U-GCN-2) of utilizing 2-hop neighbors (and $k$NN) is on average 3.37% (and 3.12%) more accurate on eight datasets. This validates that 2-hop neighbors (and $k$NN) play an important role during information propagation, especially on the networks dominated by heterophily. Furthermore, by introducing $k$NN, the derived U-GCN-3 improves performance of GCN (and U-GCN-2), i.e., on average 6.64% (and 3.32%) more accurate on eight datasets. This demonstrates that the performance of 1-hop and $k$NN neighbors can be mutually enhanced to a certain extent. In addition, U-GCN is on average 2.49% more accurate than U-GCN-3, which further validates the soundness of our new universal GCN framework that makes full use of different propagation mechanisms aiming to the network with diverse properties.

Table 3: Comparisons of our U-GCN with four variants on node classification (Percent).

| Methods | Cora | Pubm. | Cite. | Corn. | Cham. | Squi. | Wisc. | Texa. | AVG |
|---|---|---|---|---|---|---|---|---|---|
| GCN | 82.93 | 83.29 | 73.12 | 46.51 | 52.32 | 33.10 | 47.73 | 52.71 | 58.96 |
| U-GCN-1 | 74.40 | 83.92 | 68.66 | 56.59 | 48.81 | 33.84 | 64.20 | 68.22 | 62.33 |
| U-GCN-2 | 70.27 | 80.86 | 68.74 | **70.54** | 38.60 | 29.11 | 68.75 | 69.77 | 62.08 |
| U-GCN-3 | 83.67 | 81.33 | 72.79 | 65.12 | 53.51 | 34.06 | 69.89 | 62.79 | 65.40 |
| U-GCN | **84.00** | **85.22** | **74.08** | 69.77 | **54.07** | **34.39** | **69.89** | **71.72** | **67.89** |

## 6 Related Work

In line with the focus of our work, we briefly review the most related work on graph neural networks (GNNs) and homophily assumption.

**GNNs.** In recent years, GNNs have become popular for graph-based machine learning problems increasingly [13, 31]. Defferrard *et al.* [5] propose the first version of GNN by generalizing convolutional neural networks (CNNs) from regular grids (e.g., images) to irregular grids (e.g., graphs). After that comes GCN [14], a popular GNN model which obtains node embeddings by integrating high-order neighborhood information through stacked graph convolutional layers. Further, GraphSAGE [8] generalizes the aggregation beyond averaging, and models the ego-features distinctly from the neighbor-features in its subsampled neighborhood. GAT [26] introduces a node-level multi-head attention mechanisms to specify the weights from different neighborhoods.

**Homophily Assumption.** Several efforts have been made to relieve the limitation of homophily assumption, so as to improve GCNs. Geom-GCN [22] proposes a novel geometric aggregation scheme to overcome neighborhood structural information losing and long-range dependencies lacking. GPR-GNN [4] proposes a new architecture that adaptively learns the generalized pageRank (GPR) weights, to jointly optimize node feature and topological information extraction. CPGNN [34] designs a GNN framework that incorporates an interpretable compatibility matrix $H$ for modeling the homophily level. Moreover, H2GCN [35] incorporates three key designs: ego- and neighbor-embedding separation, higher-order neighborhoods and combination of intermediate, to capture information under the low level of homophily. More Recently, GGCN [32] proposes a robust and generalized model that addresses the discrepancy in features and degrees between neighbors by incorporating signed messages and learned degree corrections, so as to alleviate the homophily assumption of GCNs.

These existing methods have achieved reasonable results on handling the homophily assumption of GCNs. However, there is still a lack of insightful understanding of the key factors of a universal propagation mechanism independent of the network structural assumption, which is of great significant while often ignored by the existing GCN methods.

## 7 Discussion

In this section, we discuss what are the universal propagation mechanism in networks. Take the global spread of epidemics as an example, it is a complex, network-driven dynamic process. The combined multi-scale nature and intrinsic heterogeneity of the epidemic network make it difficult to develop an intuitive understanding of this process, to predict its time course and to locate its origin. However, Brockmann *et al.* [3] show that if we use a probabilistically motivated effective distance, rather than conventional geographic distance, to analyse propagation process, the complex spatiotemporal patterns can be simplified into simple, homogeneous wave propagation patterns. Motivated by this idea, it is of great necessary to find a universal propagation mechanism for GCNs, where universality refers to independence on homophily, heterophily or randomness network structural assumptions, so as to select the valuable nodes as neighborhoods, and meanwhile relieve the limitation of homophily assumption of GCNs. More importantly, even if there are assumptions (e.g., homophily), a universal propagation mechanism may still be able to find better and more valuable nodes as neighborhoods, rather than directly using the 1-hop network neighbors the same as the existing methods.

## 8 Conclusion

We rethink the propagation mechanism of networks with different structural properties, and surprisingly discover the segmentation rules that 1-hop, $k$NN and 2-hop neighbors are more suitable as neighborhoods in network with complete homophily, randomness and complete heterophily, respectively. However, real-world networks are complex, and may present diverse properties. We accordingly design a universal model U-GCN, which is able to select more valuable nodes as neighborhoods automatically for information propagation without relying on network structural assumptions. Empirical results on networks with different edge homophily ratio demonstrate the superiority of our new approach over some state-of-the-art methods.

Last but not least, the methods that satisfy the basic idea, that is, consider the information from nodes of 1-hop, 2-hop and $k$NN neighbors simultaneously, and at the same time, make reasonable use of these information, and fuse them effectively for the learning objectives, can be considered as general models, where our U-GCN is one of the most simple and effective methods.

## Broader Impact

In this work, we propose to analyse whether networks with different structural properties should adopt different propagation mechanisms. One interesting finding is that there are segmentation rules for the universal propagation phenomenon. Considering real-world networks are complex and may present diverse properties, a universal propagation mechanism independent of the network structural assumption have been presented. Obviously, the results of the work will have an immediate impact on improving the performance of most GCN models based on homophily assumption. Not only that,

this work will also significantly benefit applications involving network-structured data, including bioinformatics, computer vision and recommendation system.

While our research focuses on performance by designing a universal propagation mechanism on general networks with any structural properties, like many other GCNs, it provides limited explanation to its prediction. We advocate peer researchers to look into this to enhance the interpretability of modern GCNs, and make GCNs applicable in more risk-sensitive applications.

## Acknowledgements

This work is supported in by the Natural Science Foundation of China under grants 61772361, 61876128 and 62172052.

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
