# OpenReview forum: "Universal Graph Convolutional Networks"
_NeurIPS.cc/2021/Conference — NeurIPS 2021 Poster_

### Official Review · Reviewer_sxck · 2021-07-16

**Rating:** 8
**Confidence:** 5

**Summary:**

This manuscript rethinks the propagation mechanism of networks with different structural properties, and presents a novel and universal GCN model, i.e., U-GCN, for general network data. In this work, they first conduct experiments to analyze the propagation mechanism of GCNs in networks with different structural properties, and discover that there are actually segmentation rules for the propagation mechanism. They then accordingly develop a new universal GCN framework, which first introduces a multi-type convolution to extract information from different type of neighbors simultaneously, and designs a discriminative aggregation to sufficiently fuse them aiming to given learning objectives. Extensive experiments on a series of benchmark datasets demonstrate the superiority of U-GCN over some state-of-the-arts. Overall, this paper is well written, easy to follow, and of high quality.

**Limitations And Societal Impact:**

Yes, the authors demonstrate that like many other GCNs, the proposed method U-GCN provides limited explanation to its prediction.

**Main Review:**

This paper first investigates whether different networks should adopt a unified or different propagation mechanisms using several intuitive cases, and accordingly proposes a universal GCN model, namely U-GCN, for general network data.

Pros:
1. This paper is well motivated. Considering most GCNs based on the homophily assumption of networks may fail on the situations of heterophily or randomness, the authors perform motivating experiments to analyze whether different networks should adopt different propagation mechanisms.
2. The design of a universal propagation mechanism independent of the network structural assumption is a quite highlight part of this paper, which can be easily generalized to other GNNs. That is, the methods of considering the information from nodes of 1-hop network neighbors, 2-hop neighbors and kNN simultaneously, and meanwhile adaptively learning their corresponding importance for the given learning objectives can be considered as universal models.

Suggestions & Questions:
1. The author needs to modify some grammatical errors. For example:
On the 15th line: an universal -> a universal;
One the 142th line: increases -> increase, etc.
2. The authors are expected to provide some intuitive examples to demonstrate that the network dominated by a certain kind of structural properties (e.g., homophily) may contain a small amount of other structural properties (e.g., randomness), which can help the readers better understand their idea.
3. In the discriminative aggregation part, I wonder if using other aggregation operation such as concatenation instead of summation will have similar performance, what is the difference between them?
4. 2-hop neighbors or kNN may bring some technical challenges (e.g., high variance) when one attempts to scale the proposed method to large-scale graphs via selecting node pairs connected by at least two different paths for each node to set edges or constructing similarity matrix.
To sum up, the authors have made an insightful understanding in networks with different structural properties from the perspective of propagation mechanisms, and the writing is also very clear.


**Time Spent Reviewing:**

4

---

> ### Author Response · Authors · 2021-08-10
> **Reply to Reviewer sxck**
>
> Thanks a lot for your insightful comments and thorough review! The responses to your questions are listed below:
>
> **Q1:** The authors are expected to provide some intuitive examples to demonstrate that the network dominated by a certain kind of structural properties (e.g., homophily) may contain a small amount of other structural properties (e.g., randomness), which can help the readers better understand their idea.
>
> **R1:** A great suggestion. Yes, we will add more intuitive examples in the part of introduction to help the readers better understand our motivation.
>
> **Q2:** In the discriminative aggregation part, I wonder if using other aggregation operation such as concatenation instead of summation will have similar performance, what is the difference between them?
>
> **R2:** First, **in terms of feature distribution,** the summation operation is more suitable for fusing the features of the same distribution, and the concatenation operation is more suitable for adapting features with different distributions. Second, **from the perspective of model complexity,** if the features to be concatenated are of high dimension, the computation complexity of the concatenation operation will be a multiple of the summation operation.
>
> **Q3:** 2-hop neighbors or kNN may bring some technical challenges (e.g., high variance) when one attempts to scale the proposed method to large-scale graphs via selecting node pairs connected by at least two different paths for each node to set edges or constructing similarity matrix.
>
> **R3:** For 2-hop neighbors, we filter noisy nodes by limiting the number of paths connected between two nodes. In this way, we can reasonably limit the number of 2-hop neighbors, largely reducing its computational complexity.
>
> The calculation of a kNN graph typically needs $O(n^2h)$ time on a network with $n$ nodes and $h$ features. However, quick algorithms such as KD-tree [Shevtsov et al. Comput. Graph. Forum 2007] and Ball-tree [Liu et.al. J. Mach. Learn. Res. 2006] have also been proposed which reduce its complexity to $O(log_2n)$. In this case, the calculation of kNN will not increase the complexity of GNNs.

---

> > ### Comment · Reviewer_sxck · 2021-09-03
> > **Thank you for your response**
> >
> > Thank you for your response. All my concerns are addressed. I will keep my initial recommendation for acceptance.

---

### Official Review · Reviewer_TX8J · 2021-07-17

**Rating:** 6
**Confidence:** 4

**Summary:**

The paper proposes Universal GCN, an architecture built on top of GCN that utilizes 3 different neighborhoods per node to generate 3 node vectors that are then fused together via an attention mechanism. The 3 different neighborhoods are:
1. 1-hop neighborhood of the node
2. 2-hop neighborhood of the node
3. kNN based neighborhood of the node where proximity is calculated based on cosine similarity of node attributes

To motivate the need for this architecture, the authors first elaborate on phenomena like homophily, heterophily, and presence of random connections in real-world graphs to highlight that GNNs that solely rely on homophily may not perform well on real-world graphs.

**Main Review:**

Overall, this paper is an interesting one. I like the motivation presented by the authors for their work. Indeed homophily is not the only phenomenon in graphs, and graph neural networks should be able to capture other phenomena like heterophily or some any other neighborhood pattern present in a given graph. To be able to do so, we do need GNNs that have higher discriminative power than vanilla GCNs. The authors demonstrate this empirically on artificially generated graphs by comparing the attention values given to neighborhood representations generated from 1-hop, 2-hop and kNN based neighbors which show that they respectively do well on homophilic, heterophilic, and random graphs.

That said, there are several weaknesses in the paper that lie mainly in the experimentation section:
1. In table 2, are the GCN, GAT, GraphSAGE results from 2 layer networks or 1 layer network? In my opinion, we should have results from both 1 layer and 2 layers for each of those networks.
2. In lines 213-217, the authors describe a procedure to sample the graph for applying a 2-layer GCN. But why is this sampling needed in the first place because the propagation rule complexity remains the same anyway? Also, if 2-hop neighbors are directly used, why is a 2-layer GCN applied and not a 1-layer?
3. No significance testing is performed on the results to show the statistical significance of the improvements made by U-GCN over other architectures.
4. There are GNN architectures like Graph Isomorphism Networks (GIN) that have high discriminative power; but U-GCN hasn’t been compared to those.
5. Wouldn’t a better way to measure homophily be to take the average of ratio of neighbors per node within k-hops of the node that have the same label? That way, the homophily metric can be calculated for or unto any number of hops.

**Time Spent Reviewing:**

2 hours

---

> ### Author Response · Authors · 2021-08-10
> **Reply to Reviewer TX8J**
>
> Thanks a lot for your insightful comments and thorough review! The responses to your questions are listed below:
>
> **Q1:** In table 2, are the GCN, GAT, GraphSAGE results from 2 layer networks or 1 layer network? In my opinion, we should have results from both 1 layer and 2 layers for each of those networks.
>
> **R1:** The results of GCN, GAT and GraphSAGE are from 2 layers. We further compare to them with 1 layer (Table R1). As shown, U-GCN also have consistently strong performance across the full spectrum of high-middle-low homophily.
>
> ​												Table R1: Comparison to GNNs with 1 and 2 layers.
>
> | Methods   | Layers | Cora      | Pubm.     | Cite.     | Corn.     | Cham.     | Squi.     | Wisc.     | Texa.     |
> | --------- | :----: | --------- | --------- | --------- | --------- | --------- | --------- | --------- | --------- |
> | GCN       |   1    | 77.73     | 61.76     | 68.91     | 48.06     | 54.01     | 31.47     | 46.02     | 54.26     |
> |           |   2    | 82.93     | 83.29     | 73.12     | 46.51     | 52.32     | 33.10     | 47.73     | 52.71     |
> | GAT       |   1    | 80.07     | 77.10     | 71.04     | 44.19     | 52.82     | 33.34     | 47.16     | 55.81     |
> |           |   2    | 83.13     | 84.42     | 72.04     | 48.06     | 51.38     | 32.27     | 46.59     | 49.61     |
> | GraphSage |   1    | 80.80     | 51.50     | 70.20     | 54.26     | 20.74     | 29.44     | 45.45     | 54.26     |
> |           |   2    | 82.20     | 83.03     | 71.41     | 53.49     | 42.29     | 26.89     | 56.82     | 53.49     |
> | **U-GCN** |   2    | **84.00** | **85.22** | **74.08** | **69.77** | **54.07** | **34.39** | **69.89** | **71.72** |
>
>
>
> **Q2:** In lines 213-217, the authors describe a procedure to sample the graph for applying a 2-layer GCN. But why is this sampling needed in the first place because the propagation rule complexity remains the same anyway? Also, if 2-hop neighbors are directly used, why is a 2-layer GCN applied and not a 1-layer?
>
> **R2:** Real networks typically have diverse properties, e.g., a network dominated by homophily may contain a small amount of randomness or heterophily. In such cases, only certain types of neighbors may play a major role. Thus, we need to adopt several strategies to filter noise nodes for the auxiliary neighbor types, e.g., for 2 hop neighbors the number may raise exponentially with the increase of network scale. So, if there is only one path between the node and its 2-hop neighbors, we will filter out this path, and use the remaining paths for direct information propagation.
>
> In addition, this work is to find out which type of nodes are more suitable for direct information propagation. Therefore, in U-GCN, the 2-hop and kNN neighbors are nodes that should propagate information directly, analogous to 1-hop neighbors in other GNN models. Then, for 2-hop neighbors in our model, 2-layer will use 4-hop neighbors to cover more useful information in long-range distance, which is different from the indirect use of 2-hop neighbors in classical 2-layer GCNs. Take the adjacency network of English words [Newman et.al, Phys. Rev. E, 2006], where adjectives are only connected with nouns directly, as an extreme example. On node classification tasks, for any adjective, the information propagated by its 2-hop and 4-hop neighbors (which are semantic similar adjectives) is often believed more valuable than that from its 1- or 3-hop neighbors (which are nouns).
>
> **Q3:** No significance testing is performed on the results to show the statistical significance of the improvements made by U-GCN over other architectures.
>
> **R3:** The significance test that analyzes the statistical significance of improvements made by U-GCN over GCN-LPA (the second best method) *utilizing paired t-test* is reported in Table R2. As shown, for networks with strong randomness or heterophily, i.e., on Cornell, Wisconsin and Texas, our U-GCN has a breakout improvement. It also has a clear improvement on Cora and Chameleon, while a relatively modest improvement on Pubmed, Citeseer and Squirrel.
>
> ​						Table R2: Significance test between U-GCN and GCN-LPA utilizing paired t-test.
>
> | **Datasets**  | Cora   | Pubm. | Cite. | Corn.    | Cham.  | Squi. | Wisc.    | Texa.   |
> | ------------- | ------ | ----- | ----- | -------- | ------ | ----- | -------- | ------- |
> | Paired t-test | 0.0002 | 0.03  | 0.06  | 4.35e-07 | 0.0006 | 0.01  | 6.15e-08 | 4.3e-06 |
>
> **Q4:** There are GNN architectures like Graph Isomorphism Networks (GIN) that have high discriminative power; but U-GCN hasn’t been compared to those.
>
> **R4:** GIN is for graph classification while our task is node classification. So, for a feasible comparison, we reasonably utilize the node representation extraction part of GIN for node classification. We also compare two SOTA methods JK-Net [Xu et al. ICML 2018] and SSP [Izadi et al. IEEE BigData 2020]. As shown in Table R3, our U-GCN has a clear improvement compared to GIN, JK-Net and SSP. Especially on networks with complex structural properties (strong randomness or heterophily), e.g., on Cornell, Wisconsin and Texas, U-GCN have a breakout improvement.
>
> Also of note, our universal propagation mechanism can be orthogonal to many GNN models and thus can be readily incorporated into various GNNs to further improve performance.
>
> ​											Table R3: Comparison with SOTA methods including GIN.
>
> | Methods   | Cora      | Pubm.     | Cite.     | Corn.     | Cham.     | Squi.     | Wisc.     | Texa.     |
> | --------- | --------- | --------- | --------- | --------- | --------- | --------- | --------- | --------- |
> | GIN       | 66.27     | 84.96     | 65.36     | 26.36     | 53.13     | 33.10     | 39.20     | 37.21     |
> | JK-Net    | 81.27     | **86.15** | 71.74     | 52.71     | 53.95     | 33.51     | 48.30     | 51.94     |
> | SSP       | 81.08     | 79.50     | 71.13     | 55.04     | 21.87     | 19.72     | 49.37     | 55.04     |
> | **U-GCN** | **84.00** | 85.22     | **74.08** | **69.77** | **54.07** | **34.39** | **69.89** | **71.72** |
>
> **Q5:** Wouldn’t a better way to measure homophily be to take the average of ratio of neighbors per node within k-hops of the node that have the same label? That way, the homophily metric can be calculated for or unto any number of hops.
>
> **R5:** A great suggestion. We use the general definition of homophily as done in existing works while will try your new idea in the following works.

---

> > ### Comment · Reviewer_TX8J · 2021-08-28
> > **Re:**
> >
> > Thank you authors for the detailed response!
> >
> > So if the U-GCN treats 2-hop neighbors as 1-hop neighbors, hence aggregating over 4-hops with its 2-layer architecture, shouldn't we have comparison of U-GCN with 4 layer GNNs too? Capturing heterophily is indeed hard for simple GNNs like GCN, and heterophily can only be captured well with long-term dependencies. kNN based neighborhood generation is an interesting approach, but I do feel we should have comparison to deeper discriminative GNNs like 4-layer GAT and 4-layer GIN (with necessary measures to ease smoothing, e.g., DropEdge, Attention Dropout, Node Masking, etc.).
> >
> > I do like the approach of having neighbors based on kNN because that way we don't just rely on observed edges in the graph for aggregation but also latent edges based on similarity across large radii in the graph. However, this comes at an added computational cost, and hence we need to have better experimentation to be convinced that heterophily and randomness can't be captured by just having deeper discriminative GNNs.
> >
> > As a side note, could the authors please reference some papers for the definition of homophily they have used?

---

> > > ### Author Response · Authors · 2021-08-31
> > > **Reply to Reviewer TX8J**
> > >
> > > Thank you again for your insightful comments! The responses to your questions are listed below:
> > >
> > > **Q1:** So if the U-GCN treats 2-hop neighbors as 1-hop neighbors, hence aggregating over 4-hops with its 2-layer architecture, shouldn't we have comparison of U-GCN with 4 layer GNNs too? Capturing heterophily is indeed hard for simple GNNs like GCN, and heterophily can only be captured well with long-term dependencies. kNN based neighborhood generation is an interesting approach, but I do feel we should have comparison to deeper discriminative GNNs like 4-layer GAT and 4-layer GIN (with necessary measures to ease smoothing, e.g., DropEdge, Attention Dropout, Node Masking, etc.).
> > >
> > > **R1:** According to your suggestion, we have added the comparisons to 4-layer GAT and 4-layer GIN, and used DropEdge to ease smoothing. For each dataset, we randomly remove $p|E|$ of the edges and redo the normalization on the adjacency matrix before each training epoch, where $p$ is searched in the range of [0, 0.99]. As shown in Table R1, our U-GCN has a clear improvement, especially on networks with strong heterophily, e.g., on Wisconsin and Texas.
> > >
> > > This is because for networks with strong heterophily, it is important to treat 2-hop neighbors as neighborhoods for direct information propagation, rather than performing information propagation indirectly by introducing long-range dependencies as deeper discriminative GNNs. The direct propagation and indirect propagation reflect different network structural properties in network science, and thus give different results. This is also the reason why we use the 1-hop, 2-hop, and kNN neighbors as direct neighbors for propagation to reflect the general network structures (including homophily, heterophily, randomness, and/or their any types of mixtures), rather than using deeper GNN structures.
> > >
> > >  Table R1: Comparison with 4-layer GAT and 4-layer GIN.
> > >
> > > | Methods     | Cora      | Pubm.     | Cite.     | Corn.     | Cham.     | Squir.    | Wisc.     | Texa.     |
> > > | ----------- | --------- | --------- | --------- | --------- | --------- | --------- | --------- | --------- |
> > > | 4-layer GAT | 81.27     | 83.56     | 70.33     | 53.49     | 34.60     | 23.26     | 42.05     | 45.74     |
> > > | 4-layer GIN | 71.52     | 81.24     | 56.94     | 33.49     | 51.32     | 33.78     | 34.94     | 23.41     |
> > > | U-GCN       | **84.00** | **85.22** | **74.08** | **69.77** | **54.07** | **34.39** | **69.89** | **71.72** |
> > >
> > > **Q2:** I do like the approach of having neighbors based on kNN because that way we don't just rely on observed edges in the graph for aggregation but also latent edges based on similarity across large radii in the graph. However, this comes at an added computational cost, and hence we need to have better experimentation to be convinced that heterophily and randomness can't be captured by just having deeper discriminative GNNs.
> > >
> > > **R2:** Totally agreed. So, for a deeper comparison, we conduct experiments on Newman artificial networks [Girvan et al. PNAS 2002] with randomness and heterophily. The networks have 128 nodes divided into 4 classes where each node has on average $z_{in}$ edges (i.e., internal degree) connecting to nodes of the same class and $z_{out}$ edges (i.e., external degree) to nodes of other classes, and $z_{in} +z_{out} = 16$. Take two extreme cases as examples, that is, networks with complete randomness and complete heterophily (i.e., $z_{out} = 12$ and $z_{out} = 16$), respectively. As shown in Table R2, our U-GCN has a breakout improvement compared to deeper discriminative GNNs (i.e., 4-layer GAT and 4-layer GIN). This illustrates that heterophily and randomness cannot be captured well by just having deeper discriminative GNNs. But our approach works well, because we design a universal propagation mechanism independent of the network structural assumption, which can select or incorporate the most suitable neighbors (i.e., 1-hop, 2-hop and kNN neighbors) to perform direct information propagation automatically aiming to networks with different structural properties.
> > >
> > >  Table R2: Comparison to deeper discriminative GNNs.
> > >
> > > | Methods     | Networks with randomness | Networks with complete heterophily |
> > > | ----------- | ------------------------ | ---------------------------------- |
> > > | 4-layer GAT | 23.60                    | 23.60                              |
> > > | 4-layer GIN | 29.21                    | 21.35                              |
> > > | U-GCN       | **79.78**                | **78.65**                          |
> > >
> > > **Q3:** As a side note, could the authors please reference some papers for the definition of homophily they have used?
> > >
> > > **R3:**  We use the definition of homophily as done in references [1, 2].
> > >
> > > [1] Jiong Zhu, Yujun Yan, Lingxiao Zhao, Mark Heimann, Leman Akoglu, Danai Koutra. Beyond Homophily in Graph Neural Networks: Current Limitations and Effective Designs. NeurIPS, 2020.
> > >
> > > [2] Hongbin Pei, Bingzhe Wei, Kevin Chen-Chuan Chang, Yu Lei, Bo Yang. Geom-GCN: Geometric Graph Convolutional Networks. ICLR, 2020.

---

> > > > ### Comment · Reviewer_TX8J · 2021-09-03
> > > > **Re**
> > > >
> > > > Thank you authors for the response. I am increasing my score to 6.

---

### Official Review · Reviewer_v9av · 2021-08-01

**Rating:** 5
**Confidence:** 4

**Summary:**

This paper is to improve GCN by considering different hops relationship of the network. It starts with the observation that for networks with complete homophily, complete heterophily and randomness, 1-hop network, 1-hop neighbors, 2-hop neighbors and k-nearest neighbor (kNN) are more suitable as neighborhoods for information propagation for GCN, respectively. Therefore it proposed a new algorithm: U-GCN, which first individually propagates node information under different hops of graphs and then combine their results using discriminative aggregation. It compares with traditional GCN methods in small scale datasets.

**Limitations And Societal Impact:**

yes

**Main Review:**

The paper is well written and easy to understand. I think it is quite meaningful to study the information propagation under different types of graph. However I still have several concerns and suggestions as below:

1) Lots of GCN methods are using skip connections, so in theory it will consider 1 hop, 2 hops, and N hops relationships. So what is the main different between the proposed method and skip connection strategy. Any comparison with it?

2) the computation issue. I think the proposed method is quite expensive in terms of computation. As 1: it needs to for KNN graph using node features, which is known to be very expensive. 2: it needs to compute three separate GCN and then combine them together. In terms of memory and computation time, it would be at least 3 times larger/slower than GCN. Is this the reason why only small scale datasets (with # of nodes less than 20k) are tested?

3) The fair comparison is to compare different methods under the same number of parameters, as U-GCN will introduce more parameters (at least 3x more parameters) than other GCN based methods.

4) the performance of the proposed method is still far from SOTA. For example in Cora dataset, SOTA can achieve 90%+ accuracy (https://paperswithcode.com/sota/node-classification-on-cora), while the proposed method is only 84%. And the compared methods are all traditional methods with no comparison with SOTA methods.

**Time Spent Reviewing:**

3 hours

---

> ### Author Response · Authors · 2021-08-10
> **Reply to Reviewer v9av**
>
> Thanks a lot for your insightful comments and thorough review! The responses to your questions are listed below:
>
> **Q1:** Lots of GCN methods are using skip connections, so in theory it will consider 1 hop, 2 hops, and N hops relationships. So what is the main different between the proposed method and skip connection strategy. Any comparison with it?
>
> **R1:** There are two main differences between our U-GCN and GCN methods using skip connections. First, the problems to be solved are different. Our work focuses on discovering a universal propagation mechanism that may be suitable for GCNs on networks with various complicated structural properties (e.g., homophily, heterophily, randomness, and so on), and accordingly give a universal GNN framework to learn better node embeddings on more general network structures. Skip connection methods focus on analyzing whether the fixed but structure-dependent influence radius size (induced by common aggregation schemes of GCNs) really achieves the best representations for all nodes and tasks.
>
> Second, the methodologies are different. We find (via experimental observations) that for networks with complete homophily, heterophily and randomness, the 1-hop, 2-hop and kNN neighbors are more suitable as neighborhoods for direct information propagation, respectively. Based on this, we first perform direct information propagation independently utilizing 1-hop, 2-hop and kNNs, and then make a discriminative aggregation to learn out the importance of these three embeddings automatically aiming to networks with different structural properties. Skip connection methods typically adopt jumping knowledge (JK) networks to selectively exploit information from neighborhoods of differing locality, that is, selectively combines different aggregations at the last layer, i.e., the representations “jump” to the last layer.
>
> We also compare to a skip connection method JK-Net [Xu et al. ICML 2018] (Table R1). As shown, our U-GCN is on average 7.94% more accurate than JK-Net. Especially on networks with strong randomness or heterophily, e.g., on Cornell, Wisconsin and Texas, U-GCN has a breakout improvement. These results demonstrate the universality of our propagation mechanism based on segmentation rules. That is, U-GCN is well applicable to more general networks with various complicated structural properties.
>
> ​					Table R1: Comparison to a skip connection method JK-Net. “AVG” denotes on average.
>
> | **Methods** | Cora      | Pubm.     | Cite.     | Corn.     | Cham.     | Squi.     | Wisc.     | Texa.     | AVG.      |
> | ----------- | --------- | --------- | --------- | --------- | --------- | --------- | --------- | --------- | --------- |
> | JK-Net      | 81.27     | **86.15** | 71.74     | 52.71     | 53.95     | 33.51     | 48.30     | 51.94     | 59.95     |
> | **U-GCN**   | **84.00** | 85.22     | **74.08** | **69.77** | **54.07** | **34.39** | **69.89** | **71.72** | **67.89** |
>
> **Q2:** The computation issue. I think the proposed method is quite expensive in terms of computation. As 1: it needs to for KNN graph using node features, which is known to be very expensive. 2: it needs to compute three separate GCN and then combine them together. In terms of memory and computation time, it would be at least 3 times larger/slower than GCN. Is this the reason why only small scale datasets (with # of nodes less than 20k) are tested?
>
> **R2:** The calculation of a kNN graph typically needs $O(n^2h)$ time on a network with $n$ nodes and $h$ features. However, quick algorithms such as KD-tree [Shevtsov et al. Comput. Graph. Forum 2007] and Ball-tree [Liu et.al. J. Mach. Learn. Res. 2006] have also been proposed which reduce its complexity to $O(log_2n)$. In this case, the calculation of kNN will not increase the complexity of GNNs.
>
> Yes, our computational complexity is ~3 time of GCNs, i.e., $3 \ast O(ehmc)$, where $e$, $h$, $c$ are the numbers of edges, features and classes. But this is roughly a linearly increase of the complexity (since $3<<e$), so it will not limit the scope of application of GNNs.
>
> In addition, we use the same network datasets as done in most GNN works for evaluation, which have different structural properties. The selection of these datasets is for a faithful comparison rather than that we cannot deal with larger networks. But we also would like adding extra comparisons on larger networks.
>
> **Q3:** The fair comparison is to compare different methods under the same number of parameters, as U-GCN will introduce more parameters (at least 3x more parameters) than other GCN based methods.
>
> **R3:** Totally agreed. So, for a more fair comparison, we increase the number of model parameters of GCN (and GAT) by adding the latent size from 16 to 64, and keep ours unchanged. Then, we can compare the performance of different models with the same number of parameters. As shown in Table R2, the performance improvement with the increase of the number of model parameters is modest using either GCN or GAT frameworks. On the contrary, the improvement on the same (or say smaller) order of parameters increase using our framework is obvious, especially on networks with more complicated structural properties (i.e., on Cornell, Wisconsin and Texas). This illustrates that our performance improvement is mainly from the new universal propagation mechanism rather than just the increase of model parameters.
>
>  Table R2: The performance of GCN (and GAT) using different amount of model parameters, as well as our U-GCN. “#Params” denotes the amount of parameters, including constants x, y, z, m and n (y, z, m, n <<16x).
>
> | Methods   | Hidden_size |  #Params  | Cora      | Pubm.     | Cite.     | Corn.     | Cham.     | Squi.     | Wisc.     | Texa.     |
> | --------- | :---------: | :-------: | --------- | --------- | --------- | --------- | --------- | --------- | --------- | --------- |
> | GCN       |     16      |    16x    | 83.33     | 83.07     | 72.63     | 49.61     | 51.50     | 33.01     | 47.73     | 53.49     |
> |           |     32      |    32x    | 83.07     | 83.55     | 72.95     | 51.16     | 52.57     | 33.37     | 47.16     | 51.94     |
> |           |     64      |    64x    | 82.93     | 83.29     | 73.12     | 46.51     | 52.32     | 33.10     | 47.73     | 52.71     |
> | GAT       |     16      |   16x+y   | 83.53     | 82.79     | 72.12     | 53.49     | 51.94     | 32.85     | 42.62     | 54.26     |
> |           |     32      |  32x+y+m  | 83.73     | 83.29     | 71.58     | 50.39     | 53.70     | 31.04     | 47.73     | 51.16     |
> |           |     64      |  64x+y+n  | 83.13     | 84.42     | 72.04    | 48.06     | 51.38     | 32.27    | 46.59    | 49.61     |
> | **U-GCN** |   **16**    | **48x+z** | **84.13** | **85.28** | **73.54** | **68.99** | **54.01** | **33.95** | **71.02** | **69.77** |
>
> **Q4:** The performance of the proposed method is still far from SOTA. For example in Cora dataset, SOTA can achieve 90%+ accuracy (https://paperswithcode.com/sota/node-classification-on-cora), while the proposed method is only 84%. And the compared methods are all traditional methods with no comparison with SOTA methods.
>
> **R4:** We have added the comparison to SSP [Izadi et al. IEEE BigData 2020] and used their published code (https://github.com/russellizadi/ssp) as well as the default parameters. We adopt the split of datasets as done in most GCN experiments, i.e., for Cora, we use 140/300/1300 nodes for training/validation/test, respectively. As shown in Table R3, our U-GCN obviously outperforms SSP with a 13.8% improvement on average. (This is the same as that reported in Table III in their original paper with the same settings.) On the other hand, the 90.16% accuracy of SSP is only from a special setting as shown in their paper in Table V, i.e., by exploiting all labels for training excluding 500 + 500 nodes for the validation and test. In this setting, our U-GCN have a similar accuracy, i.e., 90.40%.
>
> Also of note, in Table 2 in our NeurIPS submission, U-GCN have been compared to GCN-LPA [Wang et al. arXiv:2002.06755 2020] (the top-3 SOTA method in the link you provided) and performs better.
>
> ​             Table R3: Comparison to SSP on settings of the original GCN paper. “AVG” denotes on average.
>
> | **Methods** | Cora      | Pubm.     | Cite.     | Corn.     | Cham.     | Squi.     | Wisc.     | Texa.     | AVG.      |
> | ----------- | --------- | --------- | --------- | --------- | --------- | --------- | --------- | --------- | --------- |
> | SSP         | 81.08     | 79.50     | 71.13     | 55.04     | 21.87     | 19.72     | 49.37     | 55.04     | 54.09     |
> | **U-GCN**   | **84.00** | **85.22** | **74.08** | **69.77** | **54.07** | **34.39** | **69.89** | **71.72** | **67.89** |

---

> > ### Comment · Reviewer_v9av · 2021-08-31
> > **time complexity concern**
> >
> > Thanks authors for answering my questions and conducting extra experiments, especially explaining the results for Cora and conducting experiments with equal model size. Based on that, I would like to increase my score to 5. I am still leaning towards rejection, because I am still not fully convinced by the time complexity and scalability.
> >
> > Like mentioned in the original reviews, building KNN graph is very time consuming. There could be some approximate KNN methods, but it will downgrade the final performance as well. Also the proposed method is at least 3x times slower than traditional GCN as it needs conducting on 3 different graphs. No large-scale experiments are shown, and it could be interesting to see some results comparing with some latest models from OGB https://ogb.stanford.edu/docs/nodeprop/.

---

> > > ### Author Response · Authors · 2021-09-02
> > > **Reply to time complexity concern**
> > >
> > > Thank you again for your insightful comments! The responses to your questions are listed below:
> > >
> > > **Q1:** Like mentioned in the original reviews, building KNN graph is very time consuming. There could be some approximate KNN methods, but it will downgrade the final performance as well.
> > >
> > > **R1:** Totally agreed. According to your suggestion, we have added the comparison to our approach using an approximate KNN method, i.e., Ball-tree [Liu et.al. J. Mach. Learn. Res. 2006]. Ball-tree uses hyperspheres to split data, and reduces the complexity of KNN from $O(n^2)$ to $O(log_2n)$, where $n$ is the number of nodes. As shown in Table R1, on average, U-GCN-ball-tree is slightly downgrade (~ 0.77%) the final performance of the original U-GCN. But it still has a breakout improvement (~ 8.16%) compared to GCN, especially on networks with strong randomness or heterophily, e.g., on Cornell and Wisconsin. These results demonstrate that our U-GCN can well maintain the accuracy even when using approximate KNN methods to reduce time complexity.
> > >
> > >    Table R1: Comparison to U-GCN-ball-tree. “AVG” denotes on average.
> > >
> > > | **Methods**     | Cora      | Pubm.     | Cite.     | Corn.     | Cham.     | Squi.     | Wisc.     | Texa.     | AVG.      |
> > > | --------------- | --------- | --------- | --------- | --------- | --------- | --------- | --------- | --------- | --------- |
> > > | GCN             | 82.93     | 83.29     | 73.12     | 46.51     | 52.32     | 33.10     | 47.73     | 52.71     | 58.96     |
> > > | U-GCN-ball-tree | **84.73**     | 85.20     | 73.16     | 66.67     | **58.52**     | 34.06     | 68.75     | 65.89     | 67.12     |
> > > | **U-GCN**       | 84.00 | **85.22** | **74.08** | **69.77** | 54.07 | **34.39** | **69.89** | **71.72** | **67.89** |
> > >
> > > **Q2:** Also the proposed method is at least 3x times slower than traditional GCN as it needs conducting on 3 different graphs. No large-scale experiments are shown, and it could be interesting to see some results comparing with some latest models from OGB https://ogb.stanford.edu/docs/nodeprop/.
> > >
> > > **R2:** Following your suggestion, we select larger datasets such as ogbn-arxiv from OGB to perform experiments. Since for these datasets we need to first use skip-gram models to generate embeddings for all nodes and then run GCN or U-GCN, we have not got the results yet within the so limited time. But we would like of course to share them if the deadline permits, and will add them in the final version.
> > >
> > > Instead, we also compared the running time of GCN using different hidden sizes (from 16 to 64), as well as our U-GCN on PubMed with 19,717 nodes and 44,338 edges (the largest dataset used in our NeurIPS submission). All experiments are conducted on a Linux server with GPU (GeForce GTX 1080 Ti) and CPU (Intel Xeon E5-2680). As shown in Table R2, our method is slightly slower than GCN with similar amount of parameters, while has a clear accuracy improvement. Also of note, the new approach also has the potential to further improve performance of some latest models since it can be orthogonal to most GNNs in theory.
> > >
> > >  Table R2: The running time of GCN using different hidden sizes, as well as our U-GCN on pubmed. “#Params” denotes the amount of model parameters, including constants x, y (y <<16x).
> > >
> > > | Methods   | Hidden_size | #Params     | Accuracy  | Running Time |
> > > | --------- | ----------- | ----------- | --------- | ------------ |
> > > | GCN       | 16          | 16x         | 83.07     | 34s          |
> > > |           | 32          | 32x         | 83.55     | 48s          |
> > > |           | 64          | 64x         | 83.29     | 56s          |
> > > | **U-GCN** | **16**      | **48x+y** | **85.28** | 61s          |

---

### Decision · Program_Chairs · 2021-09-27

**Decision:**

Accept (Poster)

**Comment:**

The work is motivated by the fact that GCNs are hinged on the homophily assumption, yet the real-world networks may contain heterogeneous structural properties, which requires different propagation mechanisms. As a result, the paper proposes an approach to combine 1-hop network neighbors, 2-hop neighbors and k-nearest neighbor, by using each mechanism independently and then fuse them via discriminative aggregation. The reviewers all commended the authors for studying the important topic, and at the same time had several concerns, which are mostly addressed in the rebuttal. In particular, Reviewer v9av and Reviewer TX8J are concerned with the computational complexity specifically with respect to kNN, and Reviewer TX8J also made an insightful point about whether "heterophily and randomness can't be captured by just having deeper discriminative GNNs". The authors's rebuttals addressed some of these points, although Reviewer v9av is still concerned with the computation complexity of the method on large datasets. The authors should address these further in the revision of the paper.